# Biodegradation of Polystyrene by *Galleria mellonella*: Identification of Potential Enzymes Involved in the Degradative Pathway

**DOI:** 10.3390/ijms25031576

**Published:** 2024-01-27

**Authors:** Sebastián Venegas, Carolina Alarcón, Juan Araya, Marcell Gatica, Violeta Morin, Estefanía Tarifeño-Saldivia, Elena Uribe

**Affiliations:** 1Department of Biochemistry and Molecular Biology, Faculty of Biological Sciences, University of Concepción, Concepción 4070409, Chile; svenegas@udec.cl (S.V.); caroalarcont@udec.cl (C.A.); marcgatica@udec.cl (M.G.); vmorin@udec.cl (V.M.); 2Department of Instrumental Analysis, Faculty of Pharmacy, University of Concepción, Concepción 4070409, Chile; jarayaq@udec.cl

**Keywords:** polystyrene biodegradation, *Galleria mellonella*, ATR-FTIR, MCR-ALS, LC-MS/MS

## Abstract

*Galleria mellonella* is a lepidopteran whose larval stage has shown the ability to degrade polystyrene (PS), one of the most recalcitrant plastics to biodegradation. In the present study, we fed *G. mellonella* larvae with PS for 54 days and determined candidate enzymes for its degradation. We first confirmed the biodegradation of PS by Fourier transform infrared spectroscopy- Attenuated total reflectance (FTIR-ATR) and then identified candidate enzymes in the larval gut by proteomic analysis using liquid chromatography with tandem mass spectrometry (LC-MS/MS). Two of these proteins have structural similarities to the styrene-degrading enzymes described so far. In addition, potential hydrolases, isomerases, dehydrogenases, and oxidases were identified that show little similarity to the bacterial enzymes that degrade styrene. However, their response to a diet based solely on polystyrene makes them interesting candidates as a potential new group of polystyrene-metabolizing enzymes in eukaryotes.

## 1. Introduction

Plastics are very stable synthetic polymers, while other man-made products can be easily recycled or degraded. The increase in plastic production has led to its accumulation in the environment. Today, the most widely used plastics are polyethylene (PE), polyethylene terephthalate (PET), polyvinyl chloride (PVC), polypropylene (PP), and polystyrene (PS) [1]. Among these, PS has proven to be exceptionally resistant to degradation by both natural and artificial means. In fact, not only does it have one of the lowest degradation rates in both marine and terrestrial environments [2] but, to date, no known enzyme can degrade this polymer [1]. Prokaryotes are the most studied group in terms of biodegradation, mainly because they are easy to maintain and have abundant resources [1]. However, their low degradation rates have led to the search for better options. Recently, the larvae of the insects *Tenebrio molitor*, *Zophobas atratus,* and *G. mellonella* have emerged as voracious consumers of PS. Jiang et al. 2021 demonstrated that all three species were able to ingest and degrade polystyrene into a variety of acids and alcohols [3,4]. An important advantage of these organisms is their higher rate of degradation [5] and the elimination of pre-processing steps, such as grinding, which makes them excellent alternatives for the efficient degradation of plastics. *G. mellonella*, also known as the wax moth, belongs to the family *Pyralidae* and feeds on beeswax, a complex natural polymer, destroying the hive in the process. In 2017, Bombelli et al. described, for the first time, the ability of *G. mellonella* to ingest and degrade polyethylene [5]. Although most studies have focused on polyethylene degradation, some have demonstrated that *G. mellonella* larvae can degrade PS. Lou et al. demonstrated through FTIR analysis that this organism can ingest polystyrene and induce its oxidation [6]. They also provided evidence of changes in microbial diversity in the gut after a PS diet; however, little is known about the involvement, if any, of the microbiome in the degradation of this polymer. Other studies have confirmed *G. mellonella* induced polystyrene degradation [3,7], including a recently conducted metabolomic study by Wang et al. 2022. In that work, the authors confirmed PS degradation using fluorescent polystyrene microbeads and monitoring their disappearance in the larvae’s intestine. Furthermore, they used LC-MS/MS to identify metabolites potentially derived from PS degradation, from which they proposed two different metabolic pathways [8]. Although these results demonstrate the ability of *G. mellonella* to biodegrade plastics, further studies are required to fully understand the underlying enzymatic machinery. To date, only one such study exists, which focused on the initial steps of enzyme-catalyzed PE oxidation in *G. mellonella* saliva [9]. Sanluis-Verdes et al. identified two enzymes potentially associated with the initial oxidation steps of PE: an alpha subunit of arylporin and a hexamerin. Strangely, both enzymes are phylogenetically associated with phenol oxidases, so the mechanism of action by which oxidation is catalyzed is still in doubt. In the present study, we confirmed the degradation of PS by *G. mellonella* by means of an ATR-FTIR analysis together with chemometric algorithms for pattern recognition. In addition, we tested the survival of this organism with an extended diet of PS and investigated the proteomic changes induced in the larvae’s intestine, with the objective of identifying potential enzymes involved in the biodegradation of PS.

## 2. Results

### 2.1. Degradation of Polystyrene by G. mellonella Larvae

As a first step, it was important to corroborate the degradation of polystyrene by *G. mellonella* under our current working conditions. Therefore, a feeding test of *G. mellonella* larvae with PS was carried out for 4 days. Larval feces were collected every day; then, the test was stopped by sacrificing the remaining larvae. FTIR-ATR was then used to determine its chemical composition. Since the depositions were very heterogeneous, the resulting spectra were probably a combination of the spectra of all the chemical components present in each corresponding pixel. Thus, a principal component analysis (PCA), followed by an MCR-ALS algorithm, was used to estimate the number of components and separate the original spectra into said number of “pure” spectra. The PCA of hyperspectral data of the control frass revealed the existence of a multitude of potential components; nevertheless, the initial three were enough to explain the observed variability. An MCR-ALS algorithm was then applied to the hyperspectral data using three components as its initial estimate. The results of the algorithm were then used to recreate concentration maps of each pure component (Figure 1a–c). Its infrared absorbance spectra presented several bands, including a moderate and broad alcohol (O-H) band at 3279 cm^−1^, two medium well-defined bands at 2922 cm^−1^, and 2853 cm^−1^ corresponding to alkane (C-H) stretching. A small series of bands between 1734 cm^−1^ and 1239 cm^−1^, were observed, the first of which is probably caused by carbonyl (C=O) stretching, and a strong band at 1107 cm^−1^ with a small shoulder at 1032 cm^−1^, commonly associated with different alkoxy (C-O) group vibrations. Apparently, this component corresponds mainly to carbohydrate-polysaccharide through the acute band at 1107 cm^−1^ with a shoulder. Pure component 2 of control feed larvae frass showed a very weak and broad band associated with the alcohol functional group (O-H) internal vibration centered at ~3162 cm^−1^ and a strong band at 2919 cm^−1^ followed by a moderate band at 2850 cm^−1^, both correlated to alkane group (C-H) stretching vibration. This component also had a moderate band at 1737 cm^−1^, probably caused by carbonyl (C=O) stretching, and a series of weak bands between 1464 cm^−1^ and 1167 cm^−1^, the last of which may be associated with alkoxy (C-O) group vibrations. The two C-H bands sp2 and sp3 present in most biomolecules stand out, but, here, they are in greater proportions than the rest of the bands. Furthermore, the band at 1737 cm^−1^ of carbonyl stretching, at that energy, most likely corresponds to ester-type bonds.

The last pure component of control frass, pure component 3, included a wide moderate plateau, ranging between 3270 cm^−1^ and 3000 cm^−1^, which may correspond to an alcohol (O-H)-bound either to a carboxylic acid, an alkane, or both, followed by moderate bands at 2874 cm^−1^ and 2800 cm^−1^, originating in alkane molecules. Finally, this component also showed a strong band at 1638 cm^−1^ with a moderate shoulder at 1533 cm^−1^, both normally associated with the carboxy (C=O) and nitrogen (N-H) or (C-N) of amides, respectively, after which a series of moderate bands between 1419 cm^−1^ and 780 cm^−1^ could be found, none of which could be assigned a sole responsible functional group. A very characteristic spectrum, especially due to the dominance of the amide carbonyl stretch at 1638 cm^−1^ accompanied by the amide II band at 1533 cm^−1^, was noted (see Appendix A in the Appendix A).

PCA of frass belonging to PS-fed larvae allowed the identification of three components, which explain more than 95% of the variance. The MCR-ALS algorithm then resolved the spectra and concentration maps of each of the three pure components (Figure 1d–f). Pure component 1 had a strong broad alcohol (O-H) band centered at 3270 cm^−1^, a strong alkane (C-H) band at 2919 cm^−1^ immediately followed by a moderate band of the same species at 2850 cm^−1^, a weak carbonyl (C=O) band at 1737 cm^−1^ band, and the two most common amide bands at 1635 cm^−1^ and 1536 cm^−1^, this last one as a weak shoulder. The fingerprint zone had a weak 1458 cm^−1^ band, which may correspond to the deformation of (-O-CH) or (C-C-H), and strong 1113 cm^−1^, 981 cm^−1^, and 900 cm^−1^ bands, with the first one associated to the vibration of alkoxy (C-O) and the remaining two to carbon–hydrogen deformations (C-H). This component is quite complex. It has characteristics associated with the three major biomolecules. It has a high proportion of C-H sp2 and sp3 bands (lipids), and the ester carbonyl band at 1737 cm^−1^ observed in the control appears but is less intense. It also presents the characteristic vibrations of proteins (amides) at 1635 and 1536 cm^−1^, and strong bands at 900 and 1116 cm^−1^ characteristic of polysaccharides.

Pure component 2 of PS-fed larvae frass had a complex set of bands between 3267 cm^−1^ and 2847 cm^−1^. These bands had strong to moderate intensities and peaked at 3267 cm^−1^, 3150 cm^−1^, 3135 cm^−1^, 3114 cm^−1^, 3078 cm^−1^, 3024 cm^−1^, 2913 cm^−1^, 2871 cm^−1^, and 2847 cm^−1^. Their positions may indicate the existence of alcohols (O-H), alkenes (C-H), or alkanes (C-H). Below 2000 cm^−1^, component 2 presented a small almost hidden shoulder band at 1716 cm^−1^, associated with carbonyl (C=O) groups, and strong bands at 1632 cm^−1^ and 1542 cm^−1^, normally associated with amides. These were followed by a series of moderate bands at 1491 cm^−1^, 1452 cm^−1^, 1377 cm^−1^, and 1233 cm^−1^ and two close and strong bands at 1053 cm^−1^ and 1029 cm^−1^, and a moderate band at 951 cm^−1^, which may be associated with a variety of functional groups such as alkene and alkoxy. This component is mostly protein, dominated by amide bands I and II.

PS-fed larvae frass’ last component, pure component 3, showed a weak broad band at 3150 cm^−1^, corresponding to a carboxylic acid-bound alcohol (O-H) and two weak bands at 2922 cm^−1^ and 2850 cm^−1^, most probably originating from alkane (C-H) symmetrical and asymmetrical vibrations. It also had a series of weak bands between 1737 cm^−1^ and 1461 cm^−1^, which may correspond to carbonyl (C=O) and other more complex vibrations, respectively. Finally, pure component 3 had bands for alkoxy (C-O) and alkene (C-H) vibrations at 1110 cm^−1^, 984 cm^−1^, and 900 cm^−1^. This component is mostly polysaccharide (see Appendix A in the Appendix A).

Taken together, the FTIR-ATR results point to a probable degradation of polystyrene into strongly oxidized alkanes and potentially alkenes or other aromatic species, even a possible benzene ring cleavage. Thus, our results provide important validation of previous publications on *G. mellonella*-mediated polystyrene degradation [3,5,6,8].

### 2.2. Effect of PS Feeding on Larvae Survival Rate

With the confirmation of PS degradation by *G. mellonella* through our FTIR-ATR results, a new assay was launched to establish the survival capacity of larvae in polystyrene; this bioassay was 54 days long. Of the initial 74 individuals in the control group, 47 survived during the trial, while among the PS-fed individuals, 42 out of 90 survived. Therefore, the control treatment had a higher survival rate (64%) than the PS-fed group (47%) (Figure 2A).

### 2.3. Comparative Proteomic Analysis after PS Degradation by G. mellonella

To identify the molecular machinery of *G. mellonella* larvae responsible for degrading PS, we analyzed the intestines of surviving larvae after 54 days of an exclusive PS diet. Two biological replicates were used per treatment to detect proteins potentially associated with PS degradation by LC-MS/MS. After proteomic analysis, we identified a wide variety of proteins through samples with 5412 in control 1, 5481 in control 2, 5327 in PS1, and 5461 in PS2. Of these, 4953 and 4852 were common in both replicates of control and PS treatment, respectively. Similarly, of these proteins, 4376 were shared between control and PS treatments, whereas 476 proteins appeared only on PS and 577 on control, for a total protein universe of 5429 proteins.

Functional annotation of the identified proteome revealed the existence of a wide variety of proteins in the intestine of *G. mellonella,* with an equally diverse set of GO molecular functions according to PANNZER2 annotation (Figure 2B). Hierarchical clustering of the detected proteins suggests treatment-induced protein modulation (Figure 2C). Differential expression analysis of the protein intensities confirmed that 246 proteins were, indeed, regulated by the PS diet, 167 of which were upregulated and 79 were downregulated (Figure 2D).

Of the 167 overexpressed proteins, 73 had enzymatic activity (according to Pfam), of which 26 were noted as potential hydrolases, 10 oxidoreductases, 1 isomerase, 14 translocases, 3 ligases, and 18 transferases (see Appendix A in the Appendix A). In this group, we identified four upregulated proteins that may be directly associated with polystyrene metabolism. These candidates correspond to three cytochromes P450 (A0A6J1WH61, A0A6J1 × 2I4, and A0A6J1WQ16) and a single flavin-binding monooxygenase (A0A6J1WE23). The detected flavin monooxygenase-like enzyme, A0A6J1WE23, shows low sequence identity with known one-component and two-component styrene monooxygenases (SMO) StyA2B from *Rhodococcus opacus* (21.39% sequence identity) and StyA from *Pseudomonas fluorescens* (10.09% sequence identity). Likewise, structural alignment (rigid JFATCAT) between the predicted 3D structure of the enzyme [10] and the reported structure for StyA from *Pseudomonas putida* (3IHM) [11] revealed a high root square mean deviation between corresponding atoms (RMSD) and moderate template modeling scores (TM-scores; 5.28 and 0.39, respectively), which confirm the low similarity between the proteins (see Appendix A, in the Appendix A). 

Considering that the *G. mellonella* diet is primarily composed of beeswax, a complex natural polymer, it stands to reason that some of the inherently expressed enzymatic machinery could degrade polystyrene and any potential intermediaries. Such enzymes would not show any differential expression between the control and PS treatment. Thus, the non-modulated fraction of proteins was screened for any enzyme of interest. This new candidate list included cytochromes P450 (A0A6J1WUC0, A0A6J3C2W6, A0A6J1WVU0, A0A6J1WVU4, A0A6J1WIH9, A0A6J1X0S3, A0A6J1WG08, A0A6J1WNR2, A0A6J1WPN0, A0A6J1WKC9 and A0A6J1WF11), aldehyde dehydrogenases (A0A6J1WKV6, A0A6J1WEK4, A0A6J1WXC9, A0A6J3BXC6, A0A6J3BRI2, A0A6J1WN53, and A0A6J1WG03), FAD-dependent oxidoreductases (A0A6J1WAJ1 and A0A6J1WQW7), and proteins related to phenol oxidases (A0A6J1WSG2, A0A6J1WVA9, and A0A6J1WN20), thus broadening our final candidate list to 27 oxidoreductases. Of the newly included cytochromes, P450 and FAD oxidoreductases, none showed high sequence identity to known styrene-degrading enzymes (see Appendix A, in the Appendix A). Furthermore, the different phenol oxidases had low sequence (Appendix A in the Appendix A) and structure (>8.00 RMSD and <0.20 TM-score) identity with both StyA and StyA2B.

Interestingly, four of the aldehyde dehydrogenases (A0A6J3BXC6, A0A6J3BRI2, A0A6J1WKV6, and A0A6J1WEK4) had moderate sequence identity with StyD of *Pseudomonas fluorescens* (~40%). Pairwise structural alignment to the reported structure of *Pseudomonas putida* phenylacetaldehyde dehydrogenase (4QYJ), [12] revealed that two of them, A0A6J1WKV6 and A0A6J1WEK4, had TM-scores of 0.95 for both proteins and an RMSD of 1.15 and 1.02, respectively (Figure 3), indicating the existence of important similarities between the proteins and the reported structure. Residue equivalency was also high, with 97.1% of the total residues of A0A6J1WKV6 being equivalent to those of the phenylacetaldehyde dehydrogenase and 96.3% of the total residues of A0A6J1WEK4 being equivalent to those of the same protein.

## 3. Discussion

Our results show, for the first time, that a polystyrene diet induces changes in the gut protein landscape of *G. mellonella* larvae. It is expected that these changes are directly associated with the metabolization of the polymer. Therefore, determining the degree of degradation of the compound in our specific working conditions was of vital importance.

In the PS-only 4-day trial, FTIR-ATR results indicated that larvae degraded polystyrene and styrene into strongly oxidized alkanes and potentially alkenes or aromatic species, including a possible benzene ring cleavage. The important feature is the complete disappearance of the bands at 1490 and 1600 cm^−1^ that are characteristic C=C vibrations of aromatic rings. Its absence practically rules out the presence of the benzene ring of styrenes in the feces. Our results provide important validation of previous reports of *G. mellonella*-mediated PS degradation.

The proteomic analysis of the larvae fed for only 54 days with PS showed the existence of 73 proteins upregulated with probable enzymatic activity, of which 26 were noted as potential hydrolases, 10 oxidoreductases, 1 isomerase, 14 translocases, 3 ligases, and 18 transferases (see Appendix A in the Appendix A). Oxidoreductases and isomerases are of special interest since the known pathways for styrene degradation require these types of enzymes. In fact, the best-known styrene degradation pathway, the styrene oxide pathway, requires a two-component system, consisting of an oxygenase and a reductase (SMO), an epoxide-cleaving isomerase (SOI), and an aldehyde dehydrogenase of the class III (PAD) [13]. Potential oxidoreductases include several cytochrome P450 and a flavin-linked monooxygenase. P450 with high specificity towards aromatic compounds and directly involved with the degradation of styrene has been reported [14,15,16]. Donoso et al. 2021 identified the existence of a P450 with high affinity towards 2-hydroxyphenylacetate (2-HPA), a potential intermediate in the degradation of styrene, and the ability to catalyze its transformation into homogentisate [15]. The three P450s detected in this study (Uniprot code: A0A6J1WH61, A0A6J1X2I4, and A0A6J1WQ16) show a high enough similarity with the CYP6B5, CYP6B2, and CYP9F2 proteins, respectively, to be included in their subfamilies. This is a highly diverse group whose families and subfamilies are involved in xenobiotic detoxification but differ greatly from one another in terms of sequence similarity and function [17]. Regarding the orthologs of the SOI enzyme, the *G. mellonella* proteome contained an important variety of isomerases, but none showed a significant degree of similarity to the described SOIs.

In addition, the unmodulated fraction of proteins was also analyzed in search of any larval enzyme that may participate in the degradation of styrene or polystyrene. Within this group of proteins, a wide range of different potential enzymes are highlighted, including cytochrome P450, aldehyde dehydrogenases, FAD-dependent oxidoreductases, a phenol oxidase, an arylforine alpha subunit, and a juvenile hormone-suppressible acidic protein. The last two proteins are especially interesting since a recent study linked them to the degradation of polymers. In their study, Sanluis-Verdes and his collaborators studied the possible oxidation of polyethylene (PE) by enzymes present in the saliva of *G. mellonella*. They found that two proteins, arylphorin alpha subunit and acid-suppressing juvenile hormone, were capable of inducing PE oxidation [9], both proteins being phylogenetically related to phenoloxidases. The authors proposed that these proteins could attack phenolic plastic additives, generating free radicals that would then initiate the oxidation of PE. The fact that these proteins have phenol oxidase activity and can be found in the *G. mellonella* proteome makes them important candidates for styrene oxidation. The sequence identity of A0A6J1WVA9 (arylphorine alpha subunit), A0A6J1WN20 (acidic juvenile hormone suppressible protein 1), and A0A6J1WSG2 (phenoloxidase subunit 2) to StyA and StyA2B was low (>15%). However, the alpha subunit of arylphorine-like and juvenile hormone-suppressible juvenile acid showed 100% sequence identity with that reported by Sanluis-Verdes.

The degradation of styrene also requires a step, in which phenylacetaldehyde is oxidated to phenylacetate (Figure 4A) by phenylacetaldehyde dehydrogenase, also known as PAD or StyD. This enzyme shows important structural similarities with the aldehyde dehydrogenases, seven of which were found in the *G. mellonella* proteome. Two of them (A0A6J1WKV6 and A0A6J1WK4) had significant structural and sequence similarities to Pseudomonas phenylacetaldehyde dehydrogenase (4QYJ) [12]. These two enzymes are strong candidates for a *G. mellonella* phenylacetaldehyde dehydrogenase.

According to a recent metabolomic study of PS degradation by Wang et al. 2022, they propose that in *G. mellonella* larvae, the degradation probably occurs through the styrene oxide-phenylacetic acid pathway or through the cresol pathway. None of these pathways have been described in organisms other than bacteria and fungi; the first pathway has only been proposed in a limited number of microorganisms such as *Pseudomonas fluorescens* [18] or *Exophiala jeanselmei* [14]. *G. mellonella* appears to possess all the enzymatic machinery to catalyze the formation of products associated with this pathway [8]. Figure 4A proposes some enzymes from the *G. mellonella* proteome that could fit into the styrene-phenylacetic oxide metabolic pathway proposed by Wang et al. (2022) [8]. The first step involves the decomposition of the polymer, a crucial step for PS biodegradation, yet no enzyme has been described so far. In the enzymes overexpressed in *G. mellonella,* we found three novel hydrolases (A0A6J1WP53, A0A6J1WT63, A0A6J3BTT6), which could be candidate enzymes for this function.

Alternatively, some authors [19] have postulated that PS could first be modified and then degraded. In Figure 4B, we propose that PS could undergo hydroxylations that would increase its solubility (mediated by P450 or FAD oxidoreductases); then, other enzymes would break the benzene rings (moxygenases or phenoloxidases), and, finally, it would be degraded to short aliphatic chains (could participate hydrolases), considering the enzymes identified in *G. mellonella.*

Further studies are required to establish the specific role of each *G. mellonella* enzyme in polystyrene metabolism. However, our work suggests great potential in polystyrene degradation for a selected set of proteins expressed in the intestine of *G. mellonella*.

## 4. Materials and Methods

### 4.1. Plastic Material

A sheet of commercially available expanded polystyrene (also known as polystyrene foam, “PS” henceforth) was acquired from a common retail seller. The polystyrene was thoroughly washed with 70% ethanol and stored in a clean space until further use. Just before the beginning of the assays, a small fragment was cut and cleaned with distilled water before being placed in one of the glass containers.

### 4.2. Polystyrene Degradation Assay

*G. mellonella* 4th instar larvae were acquired from Biobichos Ltda, a local retail seller. (Chillan, Chile). The arriving larvae were checked for deceased individuals and cleaned before being separated into either a control group consisting of 10 individuals fed a mixture of cereals and honey (in a 1:1 proportion) or a PS group, consisting of 10 individuals fed polystyrene only. Both groups were kept in glass recipients at 25 °C and a 24 h dark cycle for 4 days. Each container was cleaned of deceased larvae, silk (to avoid pupation), and depositions every day, with the depositions being stored at −20 °C until further use.

### 4.3. Larvae Feeding and Survival Assay

*G. mellonella* 4th instar larvae were obtained from the retail Biobichos Ltda. After checking and removing deceased larvae, the remaining larvae were separated into two treatment groups, each one in a different glass container. The control group consisting of 74 individuals was fed a mixture of cereals and honey (in a 1:1 proportion), whereas the PS group, which included 90 individuals, was fed polystyrene only. Both groups were kept at 25 °C and a 24 h dark cycle for the duration of the assay. The containers were cleaned of deceased larvae, silk (to avoid pupation), and depositions every 3 days until the end of the experiment. After 54 days, surviving larvae were frozen and stored at −80 °C until further use.

### 4.4. Fourier-Transform Infrared Microscopy (FTIR) and Chemometric Analysis

Depositions collected during the polystyrene degradation assay were dried at 37 °C overnight and then inspected with a dissection microscope before being analyzed using an FT-IR Spectrum Frontier/Spotlight 400 Microscopy System (Perkin Elmer, Shelton, CT, USA). Images were taken in attenuated total reflection sampling mode (ATR) with 1.56 µm pixel size, 6 cm^−1^ spectral resolution, 16 scans per image, and an ATR and atmospheric correction using the supplier software SpectrumIMAGE R1.7.1.0401. The resulting spectra data hypercube was then unfolded to a 2D matrix, which was then subjected to a principal component analysis (PCA) to determine the probable number of components in the samples (for the following analysis). A second chemometric analysis was carried out for spectral deconvolution to obtain the “pure” spectra and localization of the main chemical components in the image, using multivariate curve resolution with an alternating least squares algorithm (MCR-ALS). Pure algorithm was used as initialization, and a non-negativity constraint was applied in both spectra and concentration. All analyses were performed in Matlab (R2018b, Mathworks, Natick, MA, USA) using a MCR-ALS toolbox [20].

### 4.5. Larvae Dissection

Frozen larvae from the feeding and survival assay were thawed and had their intestines extracted. The dissection was carried on ice to minimize tissue damage. The larvae interior was exposed through a longitudinal cut on the dorsal side of the individual, after which both the initial (head) and terminal (anus) segments were carefully cut away. The intestine was then extracted from its place on the abdominal cavity using tweezers, cleaned of fat tissue with 3 washes of EDTA solution, and transferred to an Eppendorf tube containing protease/phosphatase inhibitors in a final concentration of 1× (#1861284, Thermo Scientific, Waltham, MA, USA). Larvae intestines were pooled in groups of 4, and triplicates were made for each of the two conditions.

### 4.6. Protein Extraction

The samples (3 PS tubes and 3 control tubes) were then freeze-dried (lyophilized) and resuspended in a solution containing 8 M urea, 25 mM ammonium bicarbonate, and a pH of 8. Homogenization of the resuspended proteins was carried out with 1 min of ultrasound (10 s pulses with 50% amplitude) in a cold bath. Debris was eliminated by a 5 min ice incubation followed by centrifugation (21,000× *g* for 10 min at 4 °C). The samples were quantified using Qubit 4 (Thermo Scientific) using a “Qubit protein assay” kit and analyzed by SDS-PAGE 12% [21].

### 4.7. LC-MS/MS Preparation

The remaining proteins were precipitated overnight at −80 °C with 5 volumes of cold acetone. After this step, the tubes were acclimated at room temperature for 10 min and centrifuged at 16,000× *g* for 5 min at 4 °C. The supernatant was discarded, and the pellet was washed 3 times with cold 80% acetone. The samples were subsequently left to dry in a rotatory concentrator. Next, the samples were resuspended in 30 uL of 8 M urea and 25 mM ammonium bicarbonate and then reduced with 20 mM DTT in 25 mM ammonium bicarbonate for 1 h at room temperature. Proteins were then alkylated with iodoacetamide at a final concentration of 20 mM in 25 mM ammonium bicarbonate for 1 h at room temperature and in darkness. After this, the samples were diluted 8 times with 25 mM ammonium bicarbonate to dilute the alkylating agent. Digestion was carried out with sequencing-grade trypsin (#V5071, Promega, Madison, WI, USA) in a 1:50 protease/protein (mass/mass) proportion at 37 °C for 16 h. After this time, 10% formic acid was added to stop the reaction (by a pH change). The samples were then washed in “Clean Up” Thermo Pierce C18 Spin Columns (#89870, ThermoFisher, Waltham, MA, USA) following the provider’s instructions. Finally, the peptides were dried in a rotatory concentrator overnight at 1000 rpm and 10 °C [22].

### 4.8. LC-MS/MS

Here, 200 ng of each sample peptide was injected in a nanoUHPLC “nano Elute” (Bruker Daltonics, Billerica, MA, USA) coupled to a “timsTOF Pro” (Bruker Daltonics) mass spectrometer. Liquid chromatography was run using an “Aurora UHPLC” (25 cm × 75 µm ID, 1.6 µm C18, IonOpticks) and a 90 min gradient of 2% to 35% of buffer “B” (0.1% Formic acid-Acetonitrile). Resulting data were collected using the “TimaControl 2.0” software (Bruker Daltonics) with 10 PASEF cycles, a mass range of 100–1700 *m*/*z*, 1500 V capillary ionization, 10 KHz time-of-flight frequency, and a resolution of 50,000 FWHM. Samples PS1, PS2, control 1, and control 2 were analyzed together during the same equipment run, whereas samples PS3 and control 3 were analyzed in a second separate run [22].

### 4.9. Proteomic Data Analysis

Protein identification was carried out using the software PEAKS Studio X+ (Bioinformatics Solutions), with a mass tolerance of 50 ppm, usage of monoisotopic mass, and ionic fragments of 0.05 mDa as parameters. Trypsin was selected as the digestion enzyme, with specific digestion mode and maximum of 2 missed cleavages per peptide. Carbamidomethylation of the cysteine, methionine oxidation, lysine acetylation, asparagine, and glutamine deamination, and carbamylation of lysine and the n-terminus, were chosen as post-translational modifications. Proteins were searched against the reported genome of *G. mellonella* [23], and an FDR estimation was calculated using a decoy database strategy. A minimum FDR of 1% and 1 unique peptide per protein were used as quality filters.

The remaining proteins were quantified using a label-free approximation (based on peptide intensity) with the DEP software (version 1.16) [24]. Missing values not appearing on all replicates of a condition were filtered, whereas those consistent between replicates were imputed using random draws from a Gaussian distribution centered around a constant value (0.01). The data were normalized using a variance stabilizing transformation approach, and both differential expression and statistical significance were estimated using the limma package. All proteins were annotated using the PANNZER2 software (version 2.0) [25], and potentially conserved domains were estimated with the HMMER software (version 2.43) against the Pfam-A database (version 35.0). Finally, protein sequence and structure alignments were carried out using Uniprot [26] and the PDB web tools [27] in conjunction with the AlphaFold Protein Structure Database [10,28].

## Figures and Tables

**Figure 1 ijms-25-01576-f001:**
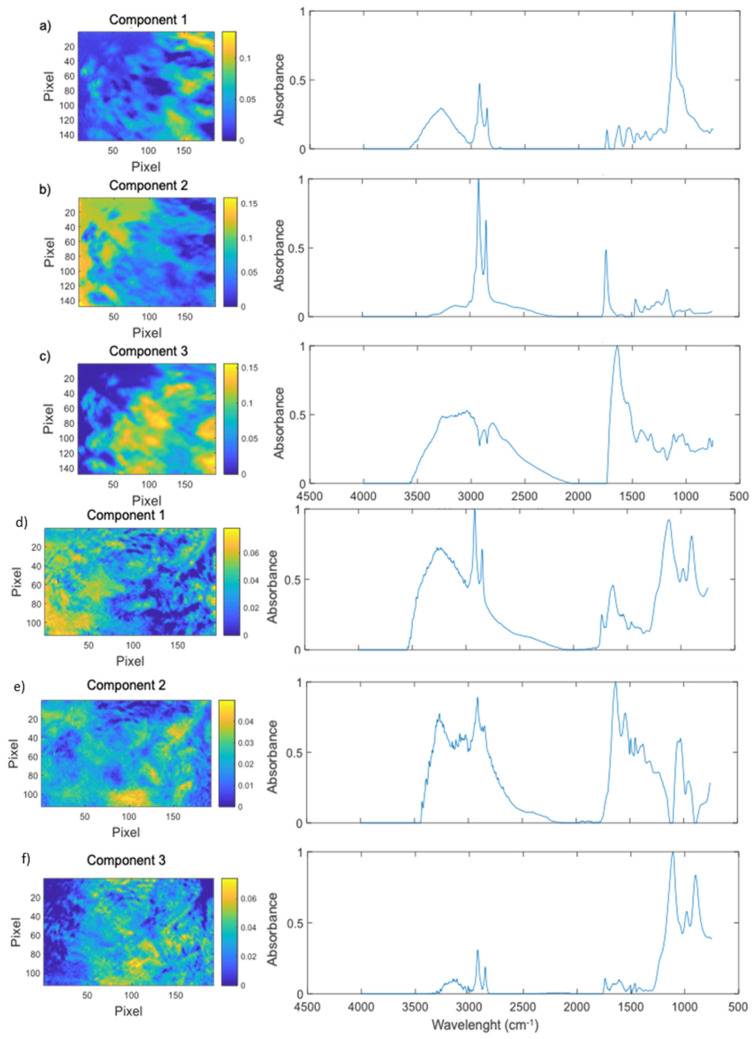
FTIR-ATR spectra and pseudo color images of the components obtained after chemometric separation of the control treatment and PS treatment samples. Control treatment: The spatial distributions of component 1 (**a**), component 2 (**b**), and component 3 (**c**) within the examined sample are shown (fit error in % (PCA) at the optimum = 14.2852. Fit error in % (exp) at the optimum = 20.7959. Percentage of explained variance (r2) at the optimum = 95.6753). PS treatment: The spatial distributions of component 1 (**d**), component 2 (**e**) and component 3 (**f**) within the examined sample are shown. Fit error in % (PCA) at the optimum = 31.9652. Fitting error in % (exp) at the optimum = 33.5266. Percentage of explained variance (r2) at the optimum = 88.7597.

**Figure 2 ijms-25-01576-f002:**
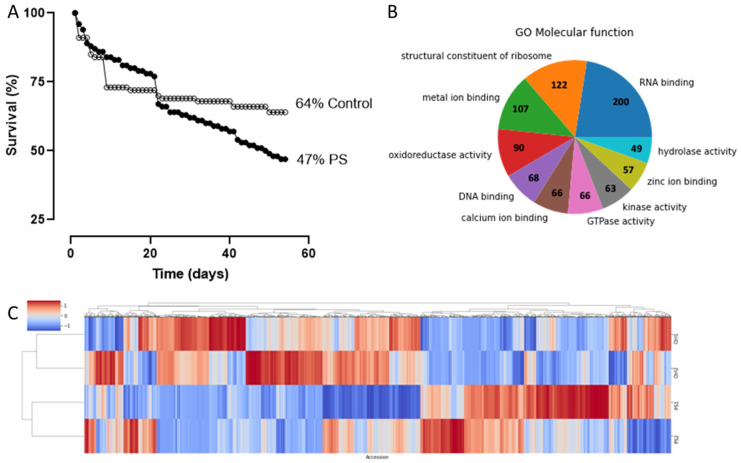
Survival rate of larvae subjected to different treatments and dynamics of the proteome induced by PS in the larval intestine. (**A**). Survival percentage of individuals in the control and PS treatment in the 54 days that the bioassay lasted. (**B**). Ten most abundant GO “Molecular Function” terms in the proteome of *G. mellonella*. (**C**). Hierarchical clustering of the 5429 proteins detected on either PS or control. (**D**). Volcano plot representing differentially modulated proteins.

**Figure 3 ijms-25-01576-f003:**
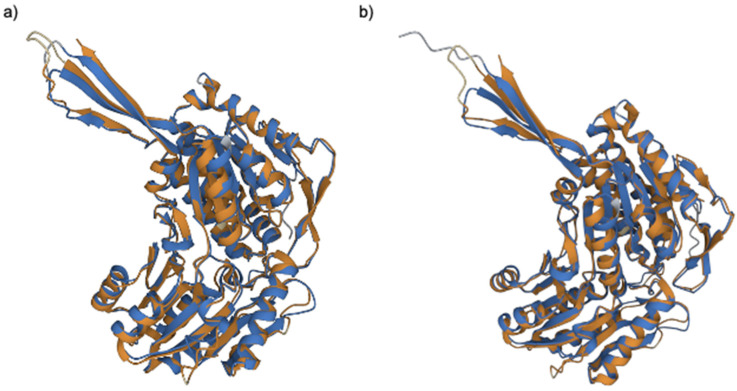
Structural alignment between *Pseudomonas putida* phenylacetaldehyde dehydrogenase crystal structure (4QYJ in orange in both images, and Alphafold predicted structures for A0A6J1WKV6 (**a**) and A0A6J1WEK4 (**b**) in blue.

**Figure 4 ijms-25-01576-f004:**
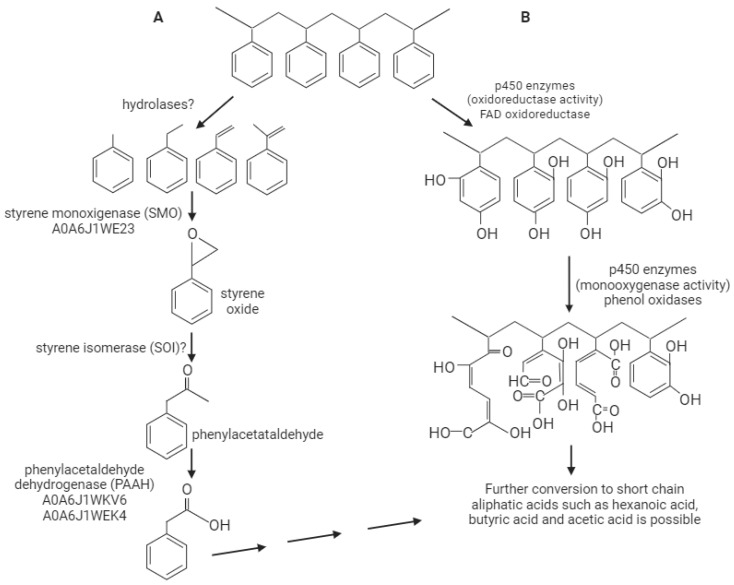
Proposed model of PS biodegradation pathways in *G. mellonella*. (**A**). The styrene oxide-phenylacetic acid pathway proposal for PS degradation for *G. melonella* with the potential candidate enzymes identified in this work (adapted from [8]). (**B**). Alternative pathway involving PS hydroxylation to increase its solubility (mediated by P450 or FAD oxidoreductases); then, other enzymes could break the benzene rings (moxygenases or phenoloxidases), and, finally, it would be degraded to short aliphatic chains (could participate hydrolases), using the enzymes of *G. mellonella*.

## Data Availability

Data is contained within the article and Appendix A.

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
