# Peer review of "Biodegradation of Polystyrene by Galleria mellonella: Identification of Potential Enzymes Involved in the Degradative Pathway"

_ijms, 2024, doi:10.3390/ijms25031576_

Round 1

Reviewer 1 Report

Comments and Suggestions for Authors

I recommend it for publication after revising some minor points listed in the attachment.

Thank you.

Author Response

Reviewer 1:

Manuscript titled “Biodegradation of Polystyrene by Galleria mellonela: Identification of Potential Enzymes  Involved in the Degradative Pathway” has been presented with interesting results about the potential  enzymes that may have a significant application in PS degradation in large-scale in the future. I recommend it  for publication after revising some minor points listed as below:

We appreciate the reviewer's comments, which are very important to improve the quality and understanding of the manuscript.

  1. Please check the instruction and correct the format of reference cited in the text and reference list also.

Dear reviewer, we appreciate your suggestion. The formatting of references throughout the manuscript was corrected.

  1. In line 33, authors mentioned the information that seems not be correct with no known enzyme can degrade PS. As the first reported PS-degrading enzyme was hydroquinone peroxidase, isolated from a lignin decolorizing strain Azotobacter beijerinckii HM121 (Nakamiya et al., 1997). And recently, there  are several studies that also investigated some enzymes such as oxidoreductases, laccases and lipases  have also been proposed to be involved in PS-degrading pathways (Hou and Majumder, 2021). Please  check and provide more correct and update information from recent researches.  

Dear reviewer, the article by Hou and Majumder, 2021, is cited in the manuscript. In this study, the authors analyzed the genome of microorganisms capable of degrading PS, and from this information predicted target enzymes that possibly degrade PS.

In the study by Nakamiya et al., 1997, the authors found that the enzyme hydroquinone peroxidase was capable of degrading polyacrylic acid, polyethylene glycol and polyvinyl alcohol, but the study did not include PS.

  1. Please add reference for all methods in section 2.

Thank you, we added references in the methodology section.

  1. Please correct all Genus/species name of microorganism strains into Italic.

Dear reviewer, genus and species names have been corrected throughout the manuscript.

Reviewer 2 Report

Comments and Suggestions for Authors

The article Biodegradation of Polystyrene by Galleria mellonella: Identification of Potential Enzymes Involved in the Degradative Pathway, authors Sebastian Venegas and co-authors, studies the degradation of polystyrene when used as a food for G. melonella larvae.

In this study, the authors attempted to determine which enzymes are involved in the process of polystyrene degradation. In general, the work is very interesting and corresponds to the profile of the IJMS.

In general, the work is informative in terms of the participation of possible enzymes in the biodegradation process. However, since a number of the enzymes identified by the authors are similar to bacterial enzymes involved in similar processes, it would be more appropriate to emphasize the role of bacteria in this process. The authors' statement about new eukaryotic enzymes involved in the degradation of polystyrene requires more substantial confirmation, in addition to the lack of a high degree of similarity of some enzymes found in this work with known bacterial ones.

However, this article presents new data regarding both potential metabolites and possible enzymes.

The MS contains a number of typos and repetitions that should be eliminated

Lines 89-90 etc. There must be a space between numbers and their units.

Lines 181-184 (They were separated…four days) repeat the corresponding section of Materials and Methods. They should be removed. The same with the repetition on Lines 275-279. This is the Methods and should not be in Result section.

Line 201, 210, 220, 234, 236  – please, check the superscript

Line 409, 461, 467, 473, 481 etc: names of bacteria should be italicized,

Line 467 “para” – also italic

The designation of cytochrome P450 should be the same throughout the text. For comparison, lines (304,326) and (427, 438, Figure 4).

Author Response

Reviewer 2:

The article Biodegradation of Polystyrene by Galleria mellonella: Identification of Potential Enzymes Involved in the Degradative Pathway, authors Sebastian Venegas and co-authors, studies the degradation of polystyrene when used as a food for G. melonella larvae.

 In this study, the authors attempted to determine which enzymes are involved in the process of polystyrene degradation. In general, the work is very interesting and corresponds to the profile of the IJMS.

In general, the work is informative in terms of the participation of possible enzymes in the biodegradation process. However, since a number of the enzymes identified by the authors are similar to bacterial enzymes involved in similar processes, it would be more appropriate to emphasize the role of bacteria in this process. The authors' statement about new eukaryotic enzymes involved in the degradation of polystyrene requires more substantial confirmation, in addition to the lack of a high degree of similarity of some enzymes found in this work with known bacterial ones.

Dear reviewer, it is important to clarify that all the sequences reported in this research belong to the G. mellonella genome. In the overexpressed enzymes from the larval intestine, the percentage of sequence identity with the bacterial enzymes is low (about 20%, see Table S3), but we found in some cases a high structural homology. The same was true for some non-overexpressed enzymes we analyzed. By this we refer to new enzymes present in G. mellonella.

However, this article presents new data regarding both potential metabolites and possible enzymes.

The MS contains a number of typos and repetitions that should be eliminated.

The MS was carefully reviewed and duplications were eliminated.

Lines 89-90 etc. There must be a space between numbers and their units.

Lines 89-90 were corrected.

Lines 181-184 (They were separated…four days) repeat the corresponding section of Materials and Methods. They should be removed. The same with the repetition on Lines 275-279. This is the Methods and should not be in Result section.

The methods sections that were in the results section will be eliminated.

Line 201, 210, 220, 234, 236  – please, check the superscript.

The indicated lines were revised and the superscripts were fixed.

Line 409, 461, 467, 473, 481 etc: names of bacteria should be italicized. Line 467 “para” – also italic.

The entire manuscript was reviewed and italicized words were corrected.

The designation of cytochrome P450 should be the same throughout the text. For comparison, lines (304,326) and (427, 438, Figure 4).

The entire manuscript was reviewed and the cytochrome P450 designation was homogenized.

Reviewer 3 Report

Comments and Suggestions for Authors

This manuscript reports on the biodegradation of polystyrene (PS) by Galleria mellonella larvae to determine candidate enzymes for the degradation. The authors used FTIR-ATR for the component identification of the degradation products, whose data are interesting. They also succeeded in identifying not a few enzymes in the larval gut by proteomic analysis to find out that potential hydrolases, isomerases, dehydrogenases, and oxidases show little similarity to the bacterial enzymes that degrade styrene.

The data shown here ought to be very important for analyzing PS-degrading enzyme series.

The paper is well documented and can be published in the present form.

Author Response

Reviewer 3:

This manuscript reports on the biodegradation of polystyrene (PS) by Galleria mellonella larvae to determine candidate enzymes for the degradation. The authors used FTIR-ATR for the component identification of the degradation products, whose data are interesting. They also succeeded in identifying not a few enzymes in the larval gut by proteomic analysis to find out that potential hydrolases, isomerases, dehydrogenases, and oxidases show little similarity to the bacterial enzymes that degrade styrene.

The data shown here ought to be very important for analyzing PS-degrading enzyme series.

The paper is well documented and can be published in the present form.

Dear reviewer, we appreciate your comments and the good consideration of our manuscript.